# Compound Structure–Composition Control on the Mechanical Properties of Selective Laser-Melted Titanium Alloys

**DOI:** 10.3390/ma15093125

**Published:** 2022-04-26

**Authors:** Guang Yang, Botao Cui, Congyu Wang, Yongdi Zhang, Chongchong Guo, Congwei Wang

**Affiliations:** College of Mechanical Engineering, Hebei University of Science and Technology, Shijiazhuang 050018, China; 15102348148@163.com (B.C.); cy_w1997@163.com (C.W.); guoccch@163.com (C.G.); wcw2451600749@126.com (C.W.)

**Keywords:** Ti6Al4V alloy, porous structure, alloy composition, selective laser melting, mechanical properties, compound control

## Abstract

In the performance optimization of the additive manufacturing of Ti6Al4V components, conventional control methods have difficulty taking into account the requirements of quality and mechanical properties of components, resulting in insufficient mechanical properties and a small control range. Therefore, combining the advantages of porous structure and alloy composition control, this paper proposed a structure–composition composite control method for selective laser-fused titanium alloy components by coupling the effects of porous structure parameters and boron content on the properties of Ti6Al4V components. Based on the Gibson–Ashby formula, the compression test of porous Ti6Al4V alloy and the tensile test of boron-containing Ti6Al4V alloy were carried out by SLM forming technology. The parameters C and *n* related to the pore parameters of porous structure were solved by the experimental data, and the analytical relationship between the pore parameters and the mechanical properties of Ti6Al4V alloy was established. The analytical relationship between boron content (*t* wt%) and mechanical properties of the alloy was established by tensile test. Finally, the Gibson–Ashby formula was used to combine the above analytical relationship, and a composite regulation model of compressive strength was obtained. The results show that the control range of the composite model ranges from 19.46–416.47 MPa, which was 45.53% higher than that obtained by controlling only pore parameters, and performance improved by 42.49%. The mechanical properties of the model are verified and the deviation between calculated values and experimental values was less than 1.3%. Taking aviation rocker arm as an example, the optimized design can improve the strength and reduce the mass of rocker arm by 51.94%. This method provides a theoretical basis for expanding the application of Ti6Al4V additive manufacturing components in aerospace and other fields.

## 1. Introduction

Titanium and its alloys are widely used in aerospace and biomedical engineering due to their non-magnetic properties, high strength, low density, and high corrosion resistance [1,2,3], and among them Ti6Al4V alloy is the most widely used. However, material redundancy is a common problem in dense titanium alloy structures, leading to a low lightweight efficiency, which can be significantly improved by designing redundant parts of a structure as porous under macroscopic pore structure control [4]. Conventional porous titanium alloy manufacturing processes include the powder metallurgy method [5], the slurry foaming method [6], and the fiber sintering method [7]. However, these methods only create random irregular-shaped pores, causing difficulties in controlling mechanical properties [8].

Selective laser melting (SLM) works based on the discrete-stacking principle and a digital model. In SLM, a high-energy laser is used as the heat source to scan and fuse powder materials layer by layer [9,10,11]. SLM has attracted extensive attention in recent years because it can freely form three-dimensional (3D) porous structures with high precision and change the mechanical properties of porous structures based on pore parameters, such as the diameter and number of struts [12]. Zaharin et al. [13] used SLM to prepare porous Ti6Al4V alloy with porosity ranging from 57.48% to 79.36%, compressive strength ranging from 10.79 MPa to 28.10 MPa, and elastic modulus ranging from 3.688 GPa to 14.586 GPa. Li et al. [14] found that when the porosity of body-centered cubic porous Ti6Al4V alloy was in the range of 88–97%, its compressive strength and elastic modulus were 0.83–6.64 MPa and 0.012–0.19 GPa, respectively. Yang et al. [15] calculated the yield strength and elastic modulus of dense Ti6Al4V alloy as 970 MPa and 113 GPa, respectively. Therefore, the existence of porosity significantly reduces the mechanical properties of Ti6Al4V alloy.

Porous titanium has the characteristics of integration of structure and function. Porous titanium integrates the advantages of titanium alloy and porous titanium. Many experts and scholars control the properties of porous titanium by adjusting the pore morphology characteristics, but this regulation method has limitations and is difficult to further meet the requirements of related fields. Luan et al. [16] asserted that the addition of a trace amount of boron could increase the nucleation rate, inhibit grain growth, and effectively improve the mechanical properties of Ti6Al4V alloy. However, simple controlling of the mechanical properties of Ti6Al4V alloy only with boron content results in limited controllability with a narrow control range, making it difficult to meet broad and diverse requirements for the mechanical properties of Ti6Al4V alloy in various applications.

Therefore, in this work, the mechanical properties of Ti6Al4V alloy prepared by SLM were characterized under the compound control of its porous structure and composition.

## 2. Materials and Methods

### 2.1. Approach

For porous metals with randomly generated and irregular pore morphology, researchers tend to abstract the random pore structure into structures with certain shapes in the study of their mechanical properties, such as the Gibson–Ashby model of cubic pores, octahedral structure model, Kelvin model of densely packed polyhedral pores, etc. In this paper, the establishment of the element structure is to discuss the connection form of the element, mainly focusing on the compressive properties of the structure, in order to study the quantitative relationship between the porosity and mechanical properties of the element, the establishment of empirical model to achieve the purpose; Gibson–Ashby empirical formula is often used to characterize the compressive strength and elastic modulus of porous structures. This empirical formula simply and clearly represents the relationship between the relative density of porous bodies and their mechanical properties, and has been widely recognized and applied in the field.

The Gibson–Ashby empirical formula is often used to characterize the compressive strength and elastic modulus of porous structures [17].
(1)$$\frac{E}{E_{s}}=C_{1}{\left(\frac{\rho }{\rho_{s}}\right)}^{n_{1}}$$
(2)$$\frac{\sigma }{\sigma_{s}}=C_{2}{\left(\frac{\rho }{\rho_{s}}\right)}^{n_{2}}$$
where *E* and *E*_s_ are the elastic moduli of porous and dense materials, respectively, *σ* and *σ*_s_ are the strengths of porous and dense materials, respectively, ρ and $\rho_{s}$ are the relative densities of porous and dense materials, respectively, *C* and *n* are constants related to the pore parameters of porous structures.

In this work, based on the Gibson–Ashby formula, the compression test of porous Ti6Al4V alloy and the tensile test of boron-containing Ti6Al4V alloy were carried out by SLM forming technology. The parameters C and *n* related to the pore parameters of porous structure were solved by the experimental data, and the analytical relationship between the pore parameters and the mechanical properties of Ti6Al4V alloy was established. The analytical relationship between boron content (*t* wt%) and mechanical properties of the alloy was established by tensile test. Finally, the Gibson–Ashby formula was used to combine the above analytical relations to obtain a composite regulation model of compressive strength, which was finally applied to aviation components, and its flowchart is presented in Figure 1.

### 2.2. Porous Structure Design

The splicing of finite basic units is the main form of porous structure. The mechanical properties of the inner diameter of the support can be obtained by the compression test of the porous structure, and the stress–strain curve can be deduced that the stress applied to the element and the porous structure tends to be the same.

Wang, H. et al. [18] pointed out that the smaller the basic unit is, the more joints the specimen is constructed, the better the mechanical properties are. Considering the requirements of bone implant aperture in the medical field and the characteristics of SLM molding process, the side length of the basic unit in this study was set as 1000 μm. Due to the limited time and capacity, this paper only studies one basic unit of various loading modes with different internal short rod numbers, rod diameters and the same rod numbers and diameters.

Chen, L. et al. [19] pointed out that the porous body with regular holes has fixed external frame but variable internal characteristics, diversified morphology, adjustable and predictable mechanical properties and pore structure attributes, and can be firmly connected. Therefore, this paper takes the regular hole as the research object to study its performance. A 3D modeling software named Solidworks was used to model the basic cell of the porous structure, which consisted of an external frame and internal struts. The external frame had a cubic structure with good three-dimensional isotropy and side length of 1000 μm (Figure 2a), and the diagonals of its six square faces were composed of semi-cylindrical struts. The internal structure was designed with struts (Figure 2b). The one end of the internal structure was mounted on the intersection of the two diagonal lines of one face of the external frame, and its other end was mounted on the adjacent face (Figure 2c). The external frame of the porous structure was fixed with firm connections, whereas the internal structure had diverse shapes, enabling tunable mechanical properties. In order to study the effect of pore parameters on the mechanical properties of porous Ti6Al4V alloy, in consideration of the actual molding, when the rod diameter is lower than 125 μm, the molding effect is poor, and when the rod diameter is higher than 200 μm, the internal residual powder is difficult to clean due to the small aperture; so, the model solved in this section is only applicable to the rod diameter range of 125–200 μm, while strut diameters were designed as 125 μm, 150 μm, 175 μm, and 200 μm. The number of internal struts ranged from 0 to 12.

### 2.3. Alloy Composition Design and Powder Preparation

#### 2.3.1. Alloy Composition Design

In this paper, a theoretical analysis of titanium alloy was made by means of microalloying. The addition of trace boron has an obvious refining effect on Ti6Al4V alloy, and TiB is generated to refine the grain of titanium alloy. Meanwhile, the hardness of TiB is much higher than that of Ti, which can improve the strength, plasticity, hardness, forming ability and fatigue performance of titanium alloy [20]. Yu et al. [21] investigated the mechanical properties of casting-formed Ti6Al4V alloy with a boron content of 0.1 wt%, and found that the strength and ductility of the alloy were improved as compared to those without boron. Sen et al. [22] found that the strength of Ti6Al4V alloy increased with the increase in the boron content from 0 wt% to 0.1 wt%. Huo et al. [23] noticed that the ductility of boron-containing Ti6Al4V alloy formed by laser cladding was continuously improved with a small loss of strength until its boron content did not exceed 0.05 wt%. However, very few reports are available on the mechanical properties of boron-containing Ti6Al4V alloy formed by SLM. Hence, in this work, 0.025 wt%, 0.05 wt%, 0.075 wt%, and 0.1 wt% of boron were added to SLM-formed Ti6Al4V alloy to comprehensively improve its mechanical properties (mass percentage of main elements in alloy powder is shown in Table 1). MDI Jade6.0 software was used to analyze the test results, so as to determine the type and structure of each phase in the sample. The final analysis results are shown in Figure 3.

It can be seen from Figure 3 that the addition of B does not change the martensite phase in SLM formed Ti6Al4V, and α’ phase exists in Ti6Al4V containing boron. When the B content was 0.025 wt% and 0.05 wt%, the TiB phase was not detected due to the low B content, while when the B content was 0.075 wt% and 0.1 wt%, with the increase in B content, the generated TiB phase also increased and appeared agglomeration phenomenon, so it was finally detected successfully.

The α and α’ phase are the same densely packed hexagonal structures, with small slip coefficient and more grain boundaries per unit area, so the strength is high. However, the dislocation plug is large, and the slip deformation is blocked, resulting in poor plasticity. TiB phase is a brittle phase, the second phase precipitation strengthening can improve its strength, but a large number of TiB phase precipitation accumulation will cause stress concentration and brittle fracture, resulting in the reduction in strength and plasticity. The analysis results from phase point of view agree with the test results of mechanical properties.

#### 2.3.2. Powder Preparation

According to the required proportion of Ti6Al4V alloy powder and pure boron powder lodeded into the loading cylinder, the loading cylinder in the active axis performs a cycle of translation, rotation and rolling composite movement, make the material along the cylinder to do circular, radial and axial three-dimensional composite movement. Thus, the mutual flow, diffusion, accumulation and doping of Ti6Al4V alloy powder and pure boron powder can be realized to achieve the effect of uniform mixing. The mixing machine used is shown in Figure 4.

### 2.4. Materials and Equipment

Grade 23 Ti6Al4V alloy powder produced by Renishaw was used in this experiment. The alloy powder parameters are shown in Table 2. The morphological characteristics of the powder are shown in Figure 5. A Renishaw AM250 Metal 3D printer, which used a high-energy laser as the heat source, was adopted for SLM, and the corresponding process parameters are listed in Table 3. In SLM, 99.99% argon was used as the protective gas to maintain the oxygen content in the build chamber below 0.1%. A C45.105 electronic universal testing machine was used to perform compression tests on porous Ti6Al4V alloy with a compression speed of 0.5 mm/min. A UTM6503 electronic universal testing machine was used to perform tensile tests on Ti6Al4V alloy with a tensile speed of 1.5 mm/min.

## 3. Results and Discussion

### 3.1. Relationship between Pore Parameters and the Mechanical Properties of Ti6Al4V Alloy

#### 3.1.1. Compression Test of Porous Ti6Al4V Alloy

The porous compression specimens prepared by SLM had a size of 10 mm × 10 mm × 10 mm (Figure 6). Compression tests were carried out to observe the variation trend of compressive properties with pore parameters. Five tests were performed for each specimen, and their average value was considered as the final result.

Figure 7a presents the compressive stress–strain curves of the specimens with zero internal struts and the strut diameters of 125–200 μm, and Figure 7b displays the compressive stress–strain curves of the specimens with 0–12 internal struts and the strut diameter of 125 μm. The porous Ti6Al4V alloy was elastoplastic; thus, its compressive stress–strain curves were divided into an elastic stage, a plateau stage, and a densification stage. In the linear elastic stage, the external frame and strut of the basic unit were elastically deformed; hence, the compression curve was a straight line with a certain slope. With the continuous increase in load, the frame and the strut started to yield. Sudden increase and decline in stress occurred at the beginning of the plateau stage due to the presence of relatively large internal pores. However, as large pores depleted, small pores were destroyed layer by layer; thus, the plateau stage reached a stable state. With further compression, solids particles were pressed to contact each other. When no pores left were left to be compressed, the alloy entered the densification stage, and the compression curve experienced a small change in strain with a sharp increase in stress.

The slope of the linear elastic stage was considered as the elastic modulus of the porous structure, and the plateau stress was calculated as the average stress in the 20–30% strain range. The mechanical properties of the alloy specimens were calculated, and their variation trend with pore parameters is plotted in Figure 8. With the continuous increase in the strut diameter and the number of internal struts, the compressive strength and elastic modulus of the specimens were in the range 18.51–295.75 MPa and 0.86–4.24 GPa respectively.

#### 3.1.2. Solution of Pore Parameters

The plateau stress and elastic modulus of porous Ti6Al4V alloy were substituted into Equations (1) and (2), respectively, to solve the pore parameters *C* and *n* when the number of internal struts ranged between 0 and 12. The value of n1 was found as 1.40, and the fitting values of other parameters are presented in Figure 9. In order to facilitate the unified characterization of the as-obtained 13 empirical formulas, the number of internal struts and their corresponding relevant parameters were again combined by the fitting method to obtain the analytical relationships between the number of internal struts and the values of $C_{1}\left(L\right)$, $C_{2}\left(L\right)$, or $n_{2}\left(L\right)$, as shown below.
(3)$$C_{1}\left(L\right)=3.796\times 10^{-4}q^{2}-1.738\times 10^{-2}q+4.483\times 10^{-1}$$
(4)$$C_{2}\left(L\right)=-4.943\times 10^{-3}q^{3}+1.172\times 10^{-1}q^{2}-9.181\times 10^{-1}q+3.752$$
(5)$$n_{2}\left(L\right)=-1.057\times 10^{-3}q^{3}+2.304\times 10^{-2}q^{2}-1.752\times 10^{-1}q+2.233$$
where q is the number of internal struts.

In Figure 9, 95% of confidence bands and prediction bands for the fitting results of $C_{1}\left(L\right)$,$C_{2}\left(L\right)$, and $n_{2}\left(L\right)$ are plotted, and the correlation coefficients $R^{2}$ of the fitting terms are listed in Table 4. All $R^{2}$ values were found to be greater than 99%, indicating that the as-obtained fitting results were accurate.

The analytical relationships between pore parameters and the mechanical properties of Ti6Al4V alloy were established by substituting $C_{1}\left(L\right)$, $C_{2}\left(L\right)$, and $n_{2}\left(L\right)$ into the Gibson–Ashby empirical formula:(6)$$E=E_{s}C_{1}\left(L\right){\left(\frac{\rho }{\rho_{s}}\right)}^{1.4}$$
(7)$$\sigma =\sigma_{s}C_{2}\left(L\right){\left(\frac{\rho }{\rho_{s}}\right)}^{n_{2}\left(L\right)}$$

### 3.2. Relationship between Boron Content and the Mechanical Properties of Ti6Al4V Alloy

#### 3.2.1. Tensile Test of Boron-Containing Ti6Al4V Alloy

The tensile specimens of boron-containing Ti6Al4V alloy formed by SLM are displayed in Figure 10. Tensile tests were carried out to analyze the variation trend of tensile properties with boron content. Three tests were performed for each specimen, and their average value was taken as the final result (Figure 11).

#### 3.2.2. Analytical Relationship between Boron Content and the Mechanical Properties of Ti6Al4V Alloy

It is noticeable from Figure 8 that when the boron content was in the range of 0–0.1 wt%, the elastic modulus of Ti6Al4V alloy was not greatly affected, whereas its tensile strength and elongation first increased and then decreased after reaching the peak value at the boron content of 0.05 wt%. Therefore, the tensile strengths of the alloy in the boron content range of 0–0.05 wt% were calculated by Equation (8).
(8)$$\sigma_{m}=-104,600t^{2}+11,630t+985$$
where $\sigma_{m}$ and t are the tensile strength of boron-containing Ti6Al4V alloy and the mass fraction of boron, respectively.

### 3.3. Establishment of Compound Control Model

The analytical relationships among pore parameters, boron content, and the mechanical properties of Ti6Al4V alloy were substituted into the Gibson–Ashby empirical formula to obtain a model that could comprehensively control the mechanical properties of the alloy, as shown below:(9)$$\sigma_{c}=\sigma_{m}C_{2}\left(L\right){\left(\frac{\rho }{\rho_{s}}\right)}^{n_{2}\left(L\right)}$$

The compound control model yielded the minimum compressive strength with 125 μm strut diameter, zero internal struts, and 0 wt% boron content and had the maximum compressive strength with 200 μm strut diameter, 12 internal struts, and 0.05 wt% boron content. These values were compared with the values obtained by only controlling pore parameters (Table 5). It is clear that the control range was expanded by 45.53%, and the controllability of mechanical properties was three times the original value.

### 3.4. Verification and Application of the Compound Control Model

#### 3.4.1. Verification of Mechanical Properties

The relationship between the alloy composition and its mechanical properties is studied on the basis of Ti6Al4V, so this paper directly on the basis of the established hole parameters, forming factors to control the performance of titanium alloy model, the method of boron content and its quantitative relationship to achieve the solution of the composite control model. The composite control model of SLM titanium alloy performance obtained by the final solution is shown in Equation (10).
(10)$$\sigma_{pl}=\sigma_{m}C_{2}{\left(\frac{\rho }{\rho_{s}}\right)}^{n_{2}}k_{2}$$

*σ_m_*—Tensile strength of SLM titanium alloy with boron content, see Equation (13); *C*_2_, *n*_2_—Related parameters of hole parameters, see Equations (11) and (12);
(11)$$C_{2}=-0.004943q^{3}+0.1172q^{2}-0.918q+3.752$$(12)$$n_{2}=-0.001057q^{3}+0.02304q^{2}-0.1752q+2.233$$

*k*_2_—Molding influence parameter, see Equation (13).

*q*—Number of internal struts, the value is an integer between 0 and 12.
(13)$$k_{1}=-4.643\times 10^{-6}p^{3}+0.002239p^{2}-0.354p+18.86$$(14)$$k_{2}=\left\{\begin{array}{c} 1,\;p\leq 150\\ -6.794\times 10^{-4}p^{2}+0.2449p-20.44,\;p>150 \end{array}\right.$$

*p*—Strut diameter (μm).

In order to verify the accuracy of the compound control model for the characterization of the mechanical properties of Ti6Al4V alloy, porous compression specimens with a boron content of 0.025%, a strut diameter of 200 μm, and the number of internal struts of 0 and 12 were prepared by SLM (Figure 12), and subsequent compression tests were performed, The bar chart of error of different internal support numbers is shown in Figure 13. The calculated values by the compound control model were compared with experimental ones (Table 6), and an error of less than 1.3% was detected, proving that the proposed model has a decent ability to characterize the compressive strength of Ti6Al4V alloy under the compound influences of pore parameters and boron content.

#### 3.4.2. Application of the Compound Control Model to Aeronautical Components

It is an important optimization goal to reduce the weight of aeronautical components without any compromise for mechanical properties [24]. The composite control of an aviation rocker arm processed by traditional casting method is carried out from the design. A casting-shaped aerial rocker arm is designed by compound control method, and the volume of the rocker arm is 47,538 mm^3^, meeting the design requirements of the maximum compressive strength of 375.73. The simplified model was imported into ANSYS for statics analysis. Hole 1 of the rocker arm was subjected to a force of 1800 N in the positive direction of *X*-axis (Figure 14a), hole 2 was subjected to a force of 2800 N in the negative direction of *Y*-axis (Figure 14b), hole 3 was a fixed constraint, and the stress cloud diagram of the aerial rocker arm with added loads is shown in Figure 15.

The maximum compressive strength of porous Ti6Al4V alloy with boron content of 0 wt% and 0.025 wt% is 292.27 MPa and 365.50 MPa, respectively, which cannot meet the requirements of mechanical properties. Therefore, in order to meet the mechanical properties, the amount of boron should be 0.05 wt%. After calculation, only when the diameter of the rod is 200 μm and the number of internal rods is 12, its compressive strength is 416.47 mpa, which can meet the strength requirements. Import Inspire software to retain the hole characteristics of the rocker arm, and then optimize the lattice structure of the rocker arm based on statics analysis for lightweight processing, as shown in Figure 16. The lightweight volume of the software porous structure is 22,849 mm^3^, reduced by 51.94% compared with the original volume. The lightweight design is realized on the premise of guaranteeing mechanical properties, which is more conducive to reducing fuel consumption and increasing the flight range of the aircraft.

## 4. Conclusions

In this work, based on the Gibson–Ashby empirical formula, the mechanical properties of Ti6Al4V alloy prepared by SLM were characterized under the compound control of its porous structure and composition. The analytical relationships among pore parameters, boron content, and the mechanical properties of the alloy were established. Further, substituting these analytical relationships into the Gibson–Ashby empirical formula, a compound control model for the mechanical properties of Ti6Al4V alloy was obtained. This compound control model tuned the compressive strength of Ti6Al4V alloy in the range of 19.46–416.47 MPa, which was 45.53% higher than that obtained by controlling only pore parameters, and performance improved by 42.49%. The available controllable points were tripled, enhancing the controllability of the alloy. The compound control model was experimentally verified with an error of less than 1.3%. Further, for a practical demonstration, the compound control model was applied to an aviation rocker arm. It was found that the model reduced the volume of the rocker arm by 51.94% without any compromise for strength requirements.

As the effect of boron on the elastic modulus of Ti6Al4V alloy was inconspicuous, the compound control model was only used to tune compressive strength. In addition, the improvement of the compressive strength of the alloy by boron was also limited. Therefore, in future studies, the effects of more elements on the mechanical properties of Ti6Al4V alloys will be investigated based on the composite regulation method presented here.

## Figures and Tables

**Figure 1 materials-15-03125-f001:**
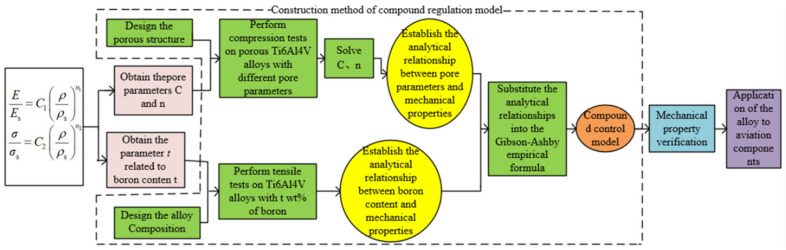
Flowchart of the compound control model.

**Figure 2 materials-15-03125-f002:**
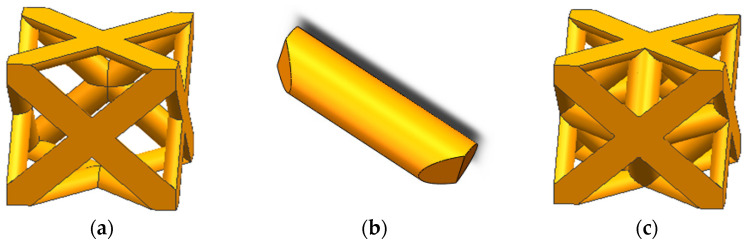
Structural model of the basic unit of the porous structure: (**a**) External frame; (**b**) internal struct; (**c**) Porous structure.

**Figure 3 materials-15-03125-f003:**
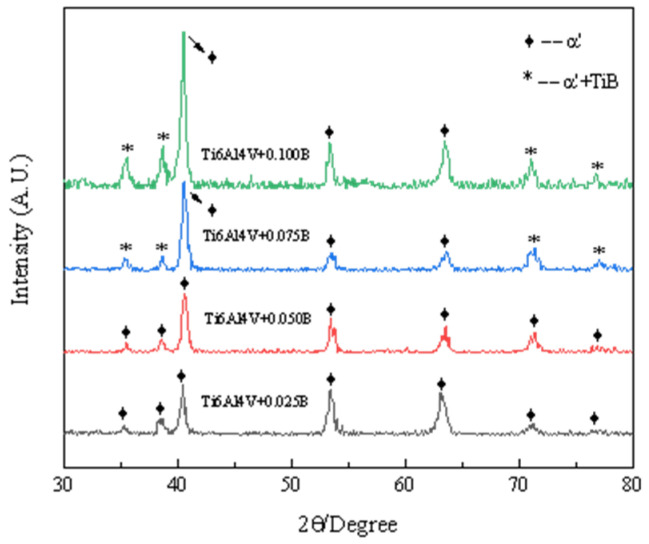
XRD analysis graph.

**Figure 4 materials-15-03125-f004:**
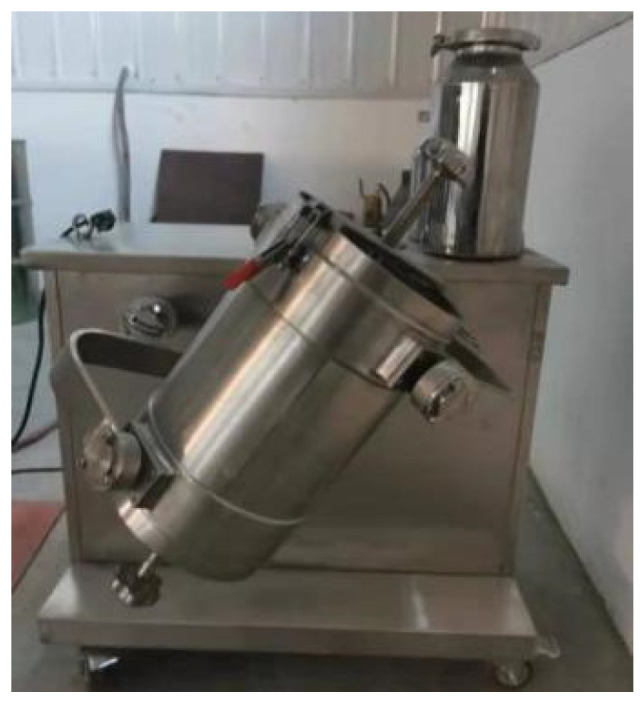
SYH-5 (10) type three-dimensional movement mixer.

**Figure 5 materials-15-03125-f005:**
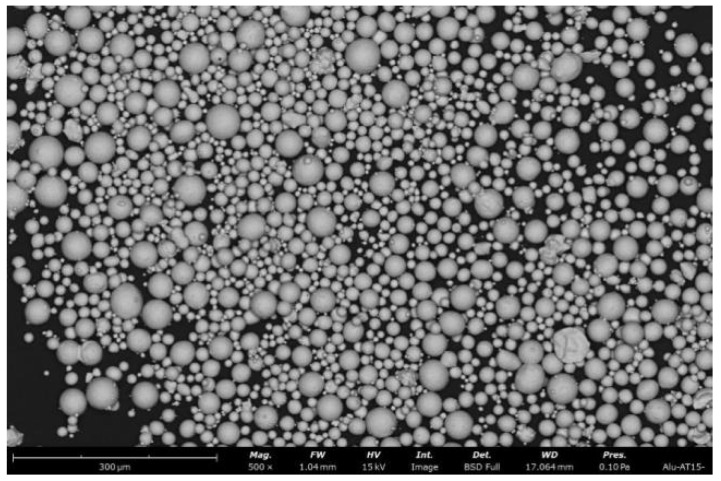
Morphology characteristics of Ti6Al4V powders.

**Figure 6 materials-15-03125-f006:**
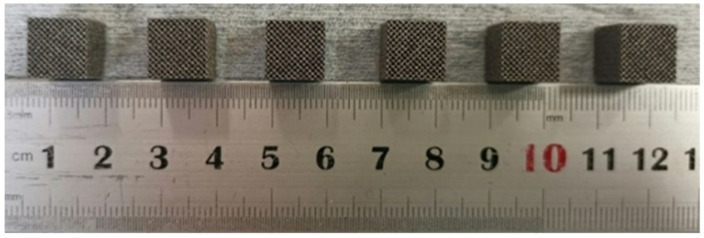
SLM-formed porous Ti6Al4V alloy compression test specimen.

**Figure 7 materials-15-03125-f007:**
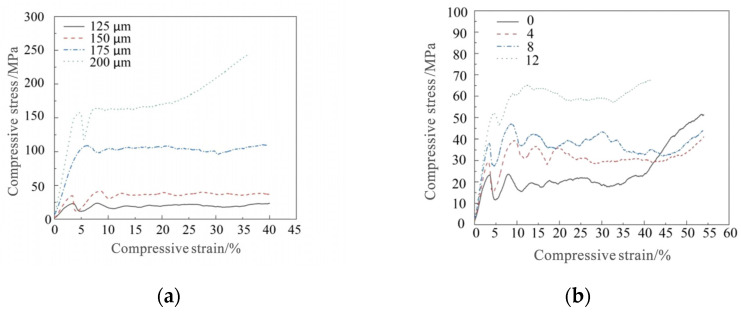
Compressive stress–strain curves: (**a**) zero internal struts and the strut diameters of 125–200 μm; (**b**) 0–12 internal struts and the strut diameter of 125 μm.

**Figure 8 materials-15-03125-f008:**
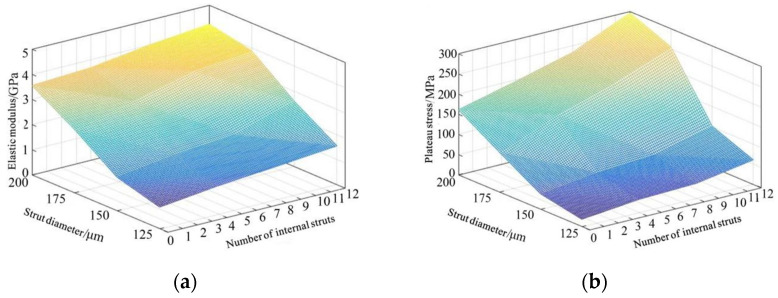
Mechanical properties as a function of strut diameter and the number of internal struts: (**a**) the variation of elastic modulus with a function of strut diameter and the number of internal struts; (**b**) the variation of platform stress with a function of strut diameter and the number of internal struts.

**Figure 9 materials-15-03125-f009:**
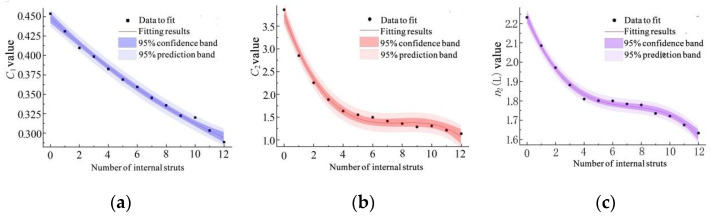
Relevant parameter values and fitting results: (**a**) $C_{1}\left(L\right)$ fitting results; (**b**) $C_{2}\left(L\right)$ fitting results; (**c**) $n_{2}\left(L\right)$ fitting results.

**Figure 10 materials-15-03125-f010:**
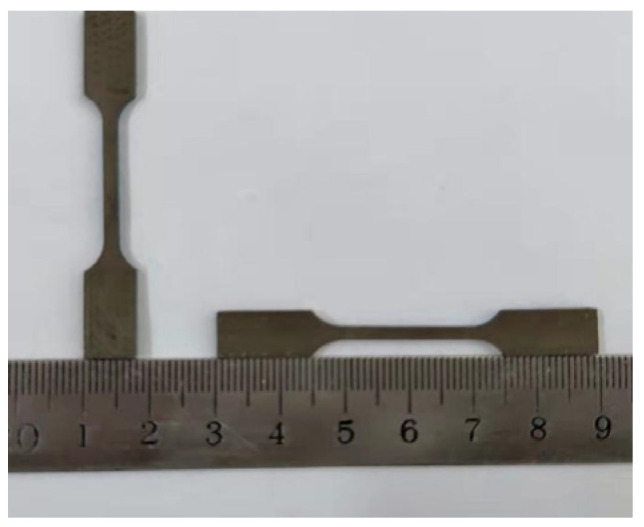
SLM-formed porous Ti6Al4V alloy tensile test specimen.

**Figure 11 materials-15-03125-f011:**
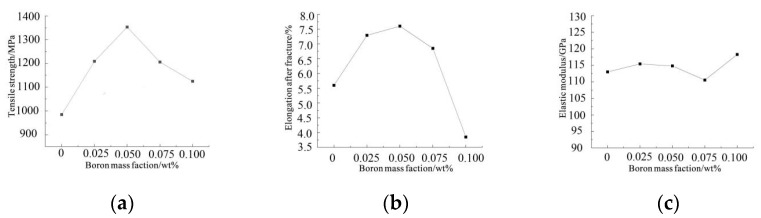
Tensile mechanical properties of SLM-formed Ti6Al4V alloys with different boron contents: (**a**) tensile strength; (**b**) elongation after fracture; (**c**) elastic modulus.

**Figure 12 materials-15-03125-f012:**
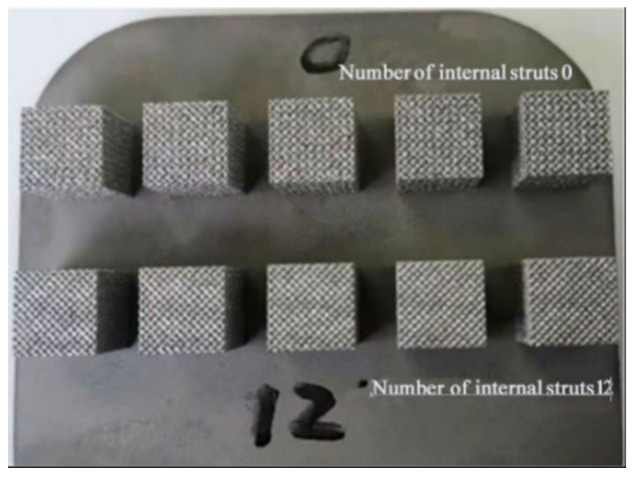
Compression test specimen of boron-containing SLM-formed porous Ti6Al4V alloy.

**Figure 13 materials-15-03125-f013:**
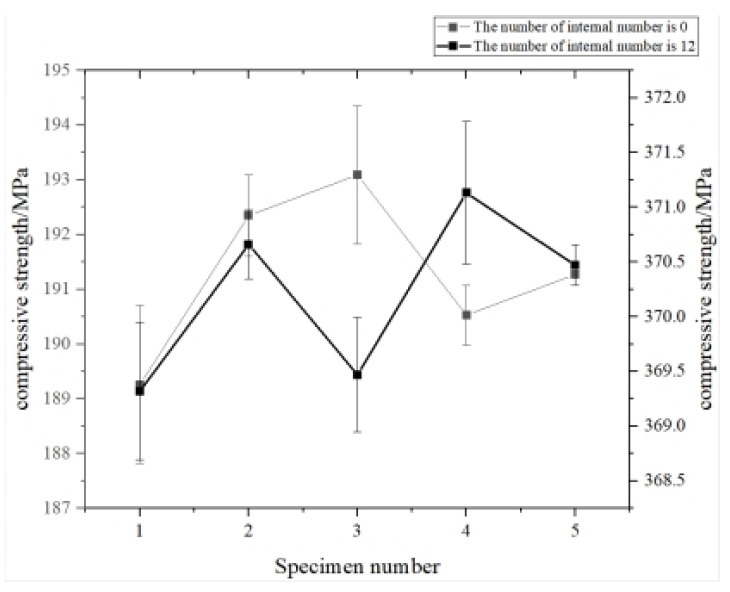
Error bars chart of compression parts error with the number of internal struts 0 and 12, respectively.

**Figure 14 materials-15-03125-f014:**
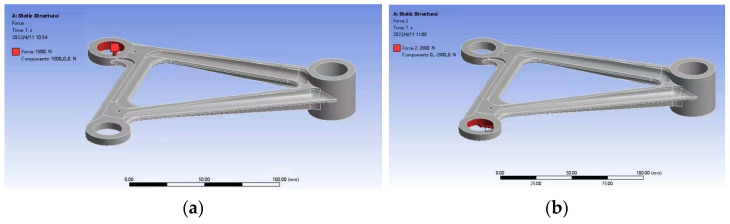
The rocker arm is loaded by force: (**a**) Load the force along the positive *X* axis; (**b**) Load the force in the negative *Y* direction.

**Figure 15 materials-15-03125-f015:**
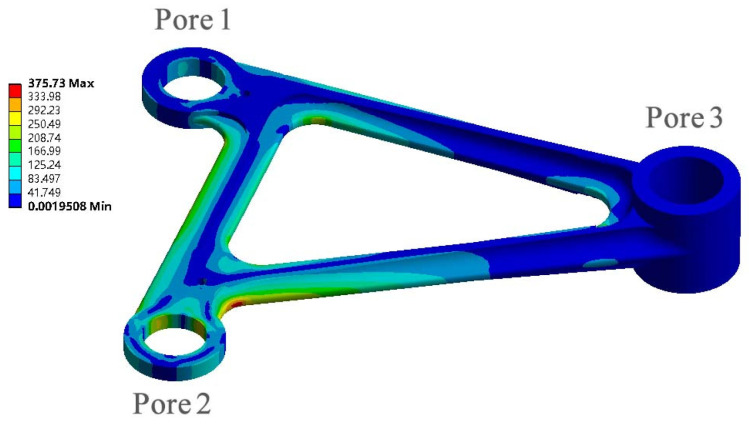
Stress nephogram of rocker arm (Pore 1).

**Figure 16 materials-15-03125-f016:**
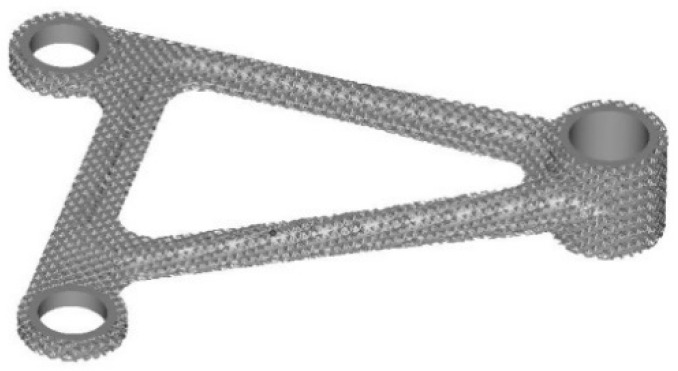
Porous rocker arm.

**Table 1 materials-15-03125-t001:** Mass percentage of main elements in alloy powder.

Elements	Al	V	Fe	B	Ti
Ti6Al4V-0.025B mass faction/%	6.49	3.91	0.17	0.025	Bal
Ti6Al4V-0.05B mass faction/%	6.51	3.85	0.17	0.050	Bal
Ti6Al4V-0.075B mass faction/%	6.57	3.74	0.17	0.075	Bal
Ti6Al4V-0.1B mass faction/%	6.68	3.61	0.17	0.100	Bal

**Table 2 materials-15-03125-t002:** Mass percentage of main elements in Ti6Al4V alloy powder.

Elements	Al	V	Fe	Ti
Ti6Al4V mass faction/%	6.48	3.96	0.17	Bal

**Table 3 materials-15-03125-t003:** SLM process parameters.

Parameter	Value
Spot diameter (μm)	75
Laser power (W)	200
Scanning speed (mm/s)	1000
Scanning pitch (μm)	67.5
Powder layer thickness (μm)	30
Build plate temp (°C)	170

**Table 4 materials-15-03125-t004:** Correlation coefficient values of fitting terms.

Fitting Term	$R^{2}$
$C_{1}\left(L\right)$	0.9945
$C_{2}\left(L\right)$	0.9913
$n_{2}\left(L\right)$	0.9945

**Table 5 materials-15-03125-t005:** Comparison of control range.

Method	Minimum Value (MPa)	Maximum Value (MPa)	Control Range (MPa)
Pore parameter control	19.46	292.27	272.81
Compound control model	19.46	416.47	397.01

**Table 6 materials-15-03125-t006:** Comparison of calculated and experimental values.

Number of Internal Struts	Calculated Value (MPa)	Experimental Value (MPa)	Error (%)
0	193.69	191.30	1.24
12	365.50	370.21	1.27

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
