# Peer review of "Compound Structure–Composition Control on the Mechanical Properties of Selective Laser-Melted Titanium Alloys"

_materials, 2022, doi:10.3390/ma15093125_

Round 1

Reviewer 1 Report

Abstract should be rewriiten by highlighting the present work

Objective of the present work with what lack in existing literature should be added above methods section

Discuss the fig 1

What modeling software was used?

On what basis, strut designs were made. Most dimensions were in microns.

Temperature specification about the equipment needs to be added

Results were not compared with the properties calculated other than SLM. Such discussion may strengthen the effectives of this work

What is the purpose of modeling part added in this work?

Necssity of Fig 10 and results were not clear

Reviewer 2 Report

  1. Figure 1 : the fonts of words in the flowchart are too small to read.
  2. Figure 4, 5 ,6,8: the images have low resolutions. Moreover, the image indices (a) and (b) must not have description after them. The description for (a) and (b) is made in the figure caption. The description after the indices makes it look like there are two figure cations.
  3. A scale bar need to be put in the images of Fig. 7.
  4.  For porous structures, there may be several mixture rules or emprical models besides the Gibson-Ashby model. It is necessary to briefly introduce these models, and then highlight the rationale of selecting Gibson-Ashby model for the present work.

Reviewer 3 Report

The manuscript "Compound structure-composition control on the mechanical properties of selective laser-melted titanium alloys" reports on the Gibson-Ashby model to control the mechanical properties of Ti6Al4V alloy prepared by selective laser melting.

From a general point of view, the topic of the manuscript is interesting and worth investigating. The manuscript is, generally, clear, well-written, well-organized, and timely. The figures are clear and appealing.

The introduction clearly states the aim of the work and sharply inserts the work within a well-focused scientific and technological framework of general interest.

The experimental approaches are clear, reliable, and strongly founded, even if the experimental section needs to be completed with some missing details. It the worth of mention is the effort by the authors in the quantitative discussions of the results concerning the analysis. On the contrary, the discussion of the results concerning the detailed analysis can be improved.

Overall, I find an interesting and valuable manuscript that requires some minor improvements before publication:

Comments:

  1. The entire article demands significant minor grammatical improvement.
  2. Please expand and improve the introduction section a little more.
  3. The introduction section is well written, whereas please refer to the latest articles.
  4. The authors should demonstrate or at least comment about scaling up difficulties or how to ease it is to scale up their technique.
  5. Please show the SEM images for powder particles used for this study.
  6. Please mention the build orientation and explain why the authors choose that.
  7. Please list the elemental compositions of Ti6Al4V alloy.
  8. Please do the XRD and confirm the crystal structure for Ti6Al4V alloy and also for boron alloys .
  9. Please write detailed captions for the figures.
  10. As the authors used pores in this article several times, please calculate the porosity also by performing XCT or SEM.
  11. Please provide the error bars for the figures.
  12. Please provide elemental compositions for the boron content samples.
  13. Please give a little more explanation on why the authors choose the boron.

Round 2

Reviewer 1 Report

Authors addressed all the queries raised, it can be considered for publication.